# A modelling approach to estimate the prevalence of treatment-resistant schizophrenia in the United States

**Michael Frank Mørup** [1]*, **Steven M. Kymes**[2], **Daniel Oudin Åström**[1]

**1** Health Economics & Epidemiology Statistics, Department of Data Science, H Lundbeck A/S, Valby, Denmark, **2** H Lundbeck Deerfield, Deerfield, IL, United States of America

* mfmq@lundbeck.com

## Abstract

### Introduction

Schizophrenia is a condition that places a significant burden on individuals with the condition, their family, and society. A large proportion of those treated for schizophrenia do not experience treatment response and are referred to as having "treatment-resistant schizophrenia" (TRS). Expert opinion has long held that the prevalence of TRS among individuals with schizophrenia is 30%, but the basis of this estimate is unclear. This article presents a model developed for estimating the prevalence of TRS in the United States 2014.

### Methods

An incidence-prevalence-mortality model was developed to estimate the prevalence of TRS in the United States. The model was populated with data from public health agencies and published literature. Prevalence in 2014 was modelled using a Markov cohort simulation for each birth cohort between 1930 to 2014.

### Results

Using different scenarios for baseline incidence, relative risks of mortality, it was estimated that approximately 22% of individuals with schizophrenia would be considered treatment-resistant in 2014.

### Discussion

The results suggests that prevalence of TRS may be somewhat lower than the 30% often reported, however this is highly dependent on the definition of treatment resistance. Methods such as this may help answer epidemiological and health policy questions as well as test the influence of key underlying assumptions.

**Data Availability Statement:** All relevant data are within the manuscript and its Supporting Information files.

**Funding:** All authors are employees of H. Lundbeck A/S. The funder provided support in the form of

salaries for authors MFM, SK, DOÅ, but did not have any additional role in the study design, data collection and analysis, decision to publish, or preparation of the manuscript. The specific roles of these authors are articulated in the 'author contributions' section.

**Competing interests:** All authors are employees of H. Lundbeck A/S. This does not alter our adherence to PLOS ONE policies on sharing data and materials.

## Introduction

Schizophrenia is one of the leading causes of disability within mental illnesses worldwide, and the debilitating disease results in a large impact on individuals and society [1]. Individuals with schizophrenia have lower life expectancy, and reduced quality of life [2, 3]. Furthermore, their relatives and caregivers also report significant impacts on daily activities [2]. Some patients diagnosed with schizophrenia only experience minimal to no symptomatic response from antipsychotic treatment and have so called treatment-resistant schizophrenia (TRS) [4–8]. The only approved pharmacotherapy for TRS is clozapine, but its use is restricted by serious adverse events and the required monitoring for patients undergoing clozapine treatment [4, 9, 10]. Additionally, half of patients with TRS are intolerant or resistant to clozapine [11].

The definition of treatment resistance is reported inconsistently across studies, and there is currently no International Statistical Classification of Diseases and Related Health Problems (ICD) code for TRS, nor does Diagnostic and Statistical Manual of Mental Disorders (DSM)-5 include a diagnostic code [12, 13]. Treatment-resistant schizophrenia is clinically defined as *a condition whereby a person with schizophrenia symptomatology has no or minimal clinical improvement following ≥2 dopamine $D_2$-preferring antipsychotics of adequate dose and duration, as assessed by a practicing clinician* [4, 14, 15].

It is described that up to 30% of all people with schizophrenia will at some point experience treatment resistance [7, 16], although numbers vary significantly in the literature, where rates of 10–60% have been reported [4, 7, 16–20]. In a Danish setting, using three different definitions of TRS, Wimberley et al (2016) reported a prevalence of 11%, 13% and 44% of patients with schizophrenia to be considered as having TRS [21], whereas 17% was reported in another study by the same author [22]. In UK, Demjaha et al (2017) reported that among a small cohort of people suffering from first episode schizophrenia, 23% were reported as treatment-resistant [23].

This discrepancy, both in the definition and prevalence of TRS in the literature makes it difficult to estimate the burden of TRS, which has important implications for decision makers in health policy and providers. An important reason for the lack of consistency in prevalence estimates is the difficulty and expense of conducting large scale epidemiological studies in this subgroup of patients with schizophrenia. One way to address this is to construct a mathematical model to simulate the temporal changes in the population of patients at risk of TRS using published epidemiological data together with national health statistics. The aim of the present study was to estimate the prevalence of TRS in the United States using a modelling approach.

## Methods

An Incidence-Prevalence-Mortality (IPM) model was used to estimate the prevalence of TRS in the United States. An IPM-model allows the investigator to estimate incidence, prevalence or mortality once two of the terms are known [24, 25]. A Markov Model assumes that an individual is always in one of several specified health states, with probabilities of transitions between the states during each cycle [26]. A one-year cycle was used for the following states in the study; Population, Schizophrenia, TRS and Dead. The population state are individuals in the cohort alive without a schizophrenia or TRS condition.

Fig 1 presents the health states in the present study.

### Data

Baseline incidence rates of schizophrenia were taken from a systematic literature review by McGrath et al (2008) [27]. Evidence suggest different incidence rates of schizophrenia based on age and sex [28]. The population under 15 years of age and above 70 years of age was not

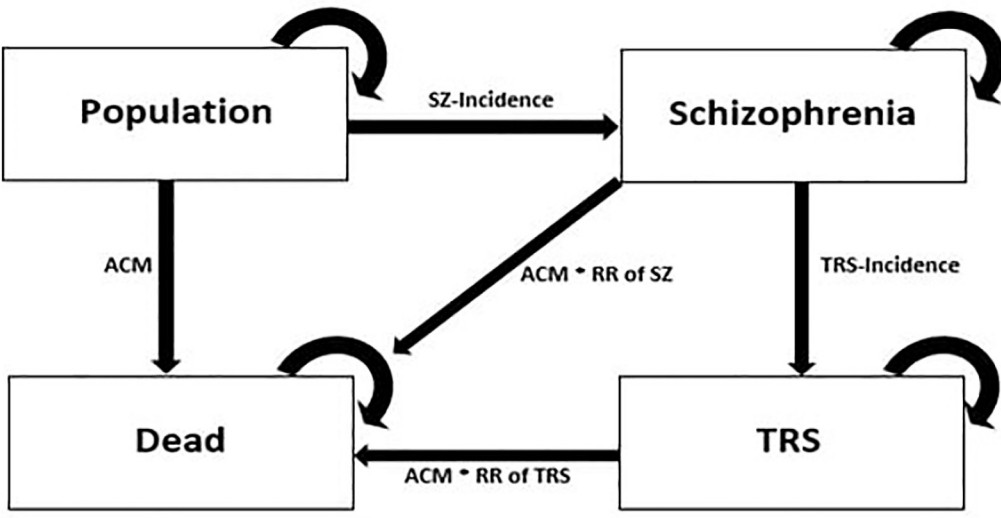

**ACM: All-cause mortality**
RR: Relative Risk
SZ: Schizophrenia
TRS: Treatment-Resistant Schizophrenia

**Fig 1. Health states in the incidence-prevalence-mortality model.**

considered to be at risk of incident schizophrenia. Considering the age and sex differences in the incidence of schizophrenia, the model input was therefore weighted with the baseline incidence rates to reflect these differences.

The incidence of TRS among patients with schizophrenia was taken from Wimberley et al 2016 [21]. They estimated the TRS incidence rate with three different proxy definitions:

1. Clozapine initiation

2. Clozapine initiation or eligible for clozapine (main definition)

3. Clozapine initiation, eligible for clozapine or 90 days' polypharmacy.

This study uses the second definition as the main definition to proxy TRS. To obtain age and sex adjusted incidence rates for TRS, the incidence was weighted to the distribution of US patients with similar proxy in the Truven Health Market Scan database for Medicaid patients [29]. The incidence rates for schizophrenia and TRS were defined as annual risk to fit the one-year cycle in the model [30].

The mortality risk of individuals with schizophrenia were based upon those found by McGrath et al and operationalized as standardized mortality rates (SMR) [27]. A recent review found continuous clozapine use to be associated with 44% lower mortality rates compared to patients continuously exposed to another antipsychotics [31]. The SMRs for TRS mortality was therefore proportionally adjusted when proxy 1 was used. The mortality risk was assumed to be similar between schizophrenia and TRS patients when proxy 2 and 3 were used. As additional analyses, the SMRs for TRS in proxy 2 were similarly adjusted.

National statistics life tables provided by the U.S Social Security Administration was used as input to the age, gender and cohort specific mortality risks [32]. US Population data for 2014 was applied from U.S. Census Bureau [33].

R version 3.6.3 was used to model TRS and R Shiny to create an application that will allow the user to generate the prevalence of TRS in the United States for different input parameters. The R-code and data used in the study are available online (S1 File).

## Statistical analysis

Applying the Markov Model assumptions with the incidence and mortality risks for each birth cohort between 1930–2014, the individual birth cohort was then modeled to 2014 for each of the four states: Population, Schizophrenia, TRS and Dead. Prevalence of TRS was calculated as the cross-section of all cohorts in 2014, as being in the TRS state and converted to prevalence per 10,000 in the US population.

The analysis was performed using the median incidence rate of schizophrenia and the median SMRs among schizophrenia patients reported in McGrath et al (2008) [27]. Furthermore, a one-way sensitivity analysis was performed for the schizophrenia incidence input and the SMR rate inputs, where the reported 10th and 90th percentile of respectively incidence and SMRs was used. In addition, the model assumed no remission from the state of TRS to Schizophrenia, and no remission between the Schizophrenia state and population state.

## Results

The prevalence estimates of the different TRS definitions, along with selected incidence rate of schizophrenia and SMRs are presented in Table 1. For each TRS definition, when the input parameters were changed, the estimated prevalence of TRS did, as expected, increase or decrease depending on scenario.

The second proxy, which was the main scenario, together with the median incidence rate and SMRs, resulted in an estimated prevalence of TRS of 6.5 per 10,000 and a proportion of

**Table 1. TRS prevalence results estimated using the IPM-model for USA 2014.**

| Input Parameters | | | Proxy 1 | | Proxy 2 | | Proxy 3 | |
|---|---|---|---|---|---|---|---|---|
| IR SZ[1] (M / F) | RR SZ[2] (M / F) | RR TRS[3] (M / F) | Prevalence per 10,000[4] (M / F / Total) | TRS proportion (%)[5] (M / F / Total) | Prevalence per 10,000 (M / F / Total) | TRS proportion (%) (M / F / Total) | Prevalence per 10,000 (M / F / Total) | TRS proportion (%) (M / F / Total) |
| **Median Incidence & Relative Risk** | | | | | | | | |
| 1.5 / 1.0 | 2.8 / 2.5 | 1.8 / 1.7 | 6.4 / 2.8 / 4.6 | 15.3 / 13.9 / 14.8 | 9.0 / 4.1 / 6.5 | 22.0 / 20.7 / 21.5 | 19.7 / 9.1 / 14.3 | 48.0 / 45.6 / 47.2 |
| **Incidence Rate Sensitivity** | | | | | | | | |
| 0.7 / 0.3 | 2.8 / 2.5 | 1.8 / 1.7 | 3.0 / 0.8 / 1.9 | 15.3 / 13.9 / 15.0 | 4.2 / 1.2 / 2.7 | 22.0 / 20.6 / 21.6 | 9.2 / 2.7 / 5.9 | 48.0 / 45.6 / 47.4 |
| 3.4 / 3.0 | 2.8 / 2.5 | 1.8 / 1.7 | 14.5 / 8.4 / 11.4 | 15.3 / 13.9 / 14.8 | 20.4 / 12.3 / 16.3 | 22.0 / 20.7 / 21.5 | 44.5 / 27.2 / 35.8 | 48.1 / 45.7 / 47.1 |
| **Relative Risk Sensitivity** | | | | | | | | |
| 1.5 / 1.0 | 1.7 / 1.5 | 1.1 / 1.0 | 7.4 / 3.1 / 5.2 | 15.7 / 14.1 / 15.2 | 10.6 / 4.6 / 7.6 | 23.0 / 21.3 / 22.5 | 23.1 / 10.2 / 16.6 | 49.9 / 46.8 / 48.9 |
| 1.5 / 1.0 | 4.7 / 5.4 | 3.1 / 3.6 | 5.2 / 2.2 / 3.7 | 14.5 / 13.3 / 14.1 | 7.2 / 3.2 / 5.2 | 20.5 / 19.2 / 20.1 | 16.0 / 7.1 / 11.5 | 45.5 / 43.0 / 44.7 |

Proxy 1 = Clozapine initiation

Proxy 2 = Clozapine initiation or eligible for clozapine. This is the main definition.

Proxy 3 = Clozapine initiation, eligible for clozapine or 90 days' polypharmacy

[1] Median Incidence Rate of schizophrenia per 10,000 for males and females. The 10th and 90th percentile incidence rate is used for sensitivity [27].

[2] Median Relative Risk of mortality in Schizophrenia state for males and females. The 10th and 90th percentile risk is used for sensitivity [27].

[3] Relative Risk of morality in TRS state for males and females (only used for Proxy 1). The TRS state in proxy 2 and 3 uses the same relative risk as the Schizophrenia state.

[4] Prevalence of Treatment-Resistant Schizophrenia per 10,000 for males, females and total.

[5] Proportion (%) of Treatment-Resistant Schizophrenia patients among people suffering from schizophrenia.

TRS of 21.5% among the schizophrenia population in the US in 2014. Comparing the one-way sensitivity analysis when changing the SMR rates while keeping the incidence rates constant, the model estimated a prevalence of TRS when using the 10th percentile of the SMR rate to be 7.6 per 10,000 as compared to 5.2 per 10,000 when the 90th percentile was used. Likewise, for the one-way sensitivity analysis of incidence rate of schizophrenia, the estimated prevalence of TRS increased from 2.7 to 16.3 per 10,000 when comparing the 10th percentile of incidence rate to the 90th percentile. Regardless of parameter input, the proportion of TRS did not change within the three different proxies.

Assuming the SMRs for schizophrenia and TRS would be different in proxy 2, as it is assumed in proxy 1, would result in a prevalence of 7.3 per 10,000 using the main definition. In contrast, if the SMR for TRS was increased by 25% relative to schizophrenia in proxy 2, it would yield a prevalence of 6.1 per 10,000.

The model found substantial impact on the estimated prevalence and the proportion of schizophrenia with TRS depending on proxy definition. The prevalence estimates ranged between 4.6 per 10,000 for the first and 14.3 per 10,000 for the third proxy. The proportion of TRS patients among patients with a schizophrenia condition ranged between 14.8 for the first proxy and 47.2 for the third proxy.

## Discussion

An IPM-model was applied to model the prevalence and proportion of TRS patients among the schizophrenia population in the US. The three proxy definitions of TRS yielded substantial differences in the estimated prevalence and proportions of TRS. Furthermore, the one-way sensitivity analysis revealed the large impact on prevalence of changing incidence and mortality risks. The model estimated the proportion of TRS patients in the schizophrenia population to be in the range of 21–22% for the main scenario by applying incidence rates and SMR rates from McGrath et al (2008) [27]. This result is similar to Demjaha et al (2017) who reported 23% to be treatment-resistant among a small UK cohort of people suffering from first episode schizophrenia [23]. The model used the same proxies for TRS as a Danish study which reported the proportion of TRS to be 11%, 13% and 44% among schizophrenia patients [21]. For each of the proxies, the model developed in this study report similar, albeit consistently higher estimates of TRS in the United States for 2014.

Despite the one-way sensitivity analyses revealing large variations in the prevalence estimates, all estimates for the different combination of proxies and parameter inputs are well within the range reported by Dammak et al 2013, who reported the proportion of TRS to be in the range of 5–60% [19]. The large differences reported in Dammak et al 2013 can mainly be attributed different definitions of TRS. The results in this study suggests that the proportion of TRS patients is driven by the definition of TRS, whereas changing risk of mortality only shows minor impact. The prevalence of TRS per 10,000 was very sensitive to changes in the incidence rate of schizophrenia, which in the main definition of TRS, changed the prevalence from 2.7 to 16.3 per 10,000 when changing the incidence rate of schizophrenia from 0.7 to 3.4 per 10,000.

The incidence rates used in this study were derived from a systematic review published in 2008 that summarized 158 studies of schizophrenia incidence [27]. By applying the 10th and 90th percentiles of the incidence rates reported in literature, plausible ranges were derived under the different assumptions. A review and meta-analysis performed on UK studies reported a pooled incidence of schizophrenia of 15.2 per 100,000 person-years [34], which is almost identical to the median incidence rate reported in McGrath et al 2008 [27]. Recently Castillejos et al (2018) reported a pooled incidence rate of 22.5 and 7.2 per 100,000 for non-

affective and affective psychosis respectively [35]. The plausible ranges coincide well with the estimated ranges by the model in this study.

The applied SMRs in the model were derived from a systematic review containing 37 studies on mortality [27]. A recent study from the US by Olfson el at (2015) [36], reported a relative risk of mortality of 3.7 from schizophrenia compared to the general population. The relative risk reported by Olfson et al (2015) is between the median and high SMR scenario in the one-way analyses, and the results in this study may thus be considered robust to changes in the SMR.

Proxy 1 assumed a lower SMR for TRS than the schizophrenia state. This assumption was also tested for the main definition proxy 2, showing a slightly lower prevalence compared to the prevalence without this assumption. Assuming a lower SMR for TRS compared schizophrenia in proxy 2 would require supportive evidence, and to the best of the authors knowledge, this does not exist.

An IPM simulation approach takes cohort time-varying mortality into account, and given reasonable inputs, it allows to test the implications of different TRS definitions and its prevalence. Informed estimates of the burden of TRS on individuals and society remains a challenging task. The results may help policymakers identify a large and tangible patient population with need for adequate treatment, and in the future avoid under -or over estimations of the societal burden of TRS.

Further research could be done to incorporate a probabilistic sensitivity analysis to the IPM model, by running the model with e.g. 10,000 draws from distributions of each input parameter, to provide a more advanced exploration of uncertainty from the inputs in the IPM model.

## Limitations

The model assumed the incidences rates and relative risks of mortality to be constant over time, despite evidence suggesting these parameters to evolve over time. Increasing incidence rates for schizophrenia was reported for those aged below 33, whereas decreasing incidence rates were reported for those aged 33 and above over the period 2000–2012 in Denmark. When analyzed together, an increase of approximately 20% was reported [37]. Changing relative risks were reported in Saha et al [38]. In addition, divergent trends in longevity have been reported; increasing longevity for the general population and decreasing longevity for the schizophrenia population [39].

The incidence rate of TRS is applied from a Danish cohort study by Wimberley et al. [21], hence the analysis is a synthesis of US and non-US sources to estimate the TRS prevalence in the US. If the incidence rates of TRS differs significantly between Denmark and the US, given the same proxy was applied, it would introduce bias to the estimate due to lack of exchangeability coming from demographic, biological composition or clinical differences. However, the clozapine use between Denmark and the US was found to be close on average, with differences depending on whether the US patients are privately or publicly insured [40]. This supports the validity of proxy one, but it was not possible to validate the main proxy.

The analysis used three different proxies of TRS, all suggested in the same article by Wimberley et al 2016 [21]. The analysis may thus have omitted other definitions of TRS available in the literature. In the present study, the main driver of the prevalence of TRS was the proxy used to define TRS, thus an additional definition would be expected to impact the estimate of TRS. To the best of the authors knowledge, there are no external data sources available allowing for external validation of our predicted TRS prevalence estimates in a representative sample.

The model used in the present study did not allow remission from the state of TRS to schizophrenia, or remission between the schizophrenia state and population state. The Remission in Schizophrenia Working Group defined consensus-based operational criteria for symptomatic remission among patients with schizophrenia [41]. However, as with the definition of TRS, there is a lack of consensus on the definition of functional remission [42]. A recent review of remission rates in schizophrenia reported rates between 17% to 78% for first-episode schizophrenia and between 16% to 62% in multiple-episode patients [43]. Despite this, it would be of interest if future IPM-models could incorporate remission.

The data used in the present study, national statistics life tables and population data, were provided by official population registers, thus the accuracy and reliability should be high.

Furthermore, the model used in the present study does not explicitly consider other factors known to impact the incidence of schizophrenia, such as ethnicity [44], migrant status [45, 46] and use of cannabis [47, 48]. Increased granularity in the input parameters of the model with regards to, for instance, differences in incidence and longevity between different ethnic groups would provide improved estimates.

## Conclusion

The study estimated that approximately 22% of the schizophrenia patients had TRS in the United States for the main scenario. The estimates are highly dependent on the proxy used and as no uniform definition of TRS is available, estimating the burden of TRS on individuals and society remains a challenging task.

## Supporting information

**S1 File.**
(ZIP)

## Author Contributions

**Conceptualization:** Michael Frank Mørup, Steven M. Kymes, Daniel Oudin Åström.

**Data curation:** Michael Frank Mørup, Daniel Oudin Åström.

**Formal analysis:** Michael Frank Mørup, Daniel Oudin Åström.

**Methodology:** Michael Frank Mørup, Steven M. Kymes, Daniel Oudin Åström.

**Project administration:** Daniel Oudin Åström.

**Validation:** Steven M. Kymes, Daniel Oudin Åström.

**Writing – original draft:** Daniel Oudin Åström.

**Writing – review & editing:** Michael Frank Mørup, Steven M. Kymes, Daniel Oudin Åström.

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
