## [Decision Letter · Decision Letter 0]

12 Feb 2020

PONE-D-19-32237

A modelling approach to estimate the prevalence of treatment-resistant schizophrenia in the United States

PLOS ONE

Dear Mr. Mørup,

Thank you for submitting your manuscript to PLOS ONE. After careful consideration, we feel that it has merit but does not fully meet PLOS ONE’s publication criteria as it currently stands. Therefore, we invite you to submit a revised version of the manuscript that addresses the points raised during the review process.

We would appreciate receiving your revised manuscript by Mar 28 2020 11:59PM. To enhance the reproducibility of your results, we recommend that if applicable you deposit your laboratory protocols in protocols.io, where a protocol can be assigned its own identifier (DOI) such that it can be cited independently in the future. For instructions see: http://journals.plos.org/plosone/s/submission-guidelines#loc-laboratory-protocols

We look forward to receiving your revised manuscript.

Kind regards,

Giuseppe Carrà, MD, PhD

Academic Editor

PLOS ONE

Journal Requirements:

Please provide an amended Funding Statement that declares *all* the funding or sources of support received during this specific study (whether external or internal to your organization) as detailed online in our guide for authors at http://journals.plos.org/plosone/s/submit-now.  Please state what role the funders took in the study.  If any authors received a salary from any of your funders, please state which authors and which funder. If the funders had no role, please state: "The funders had no role in study design, data collection and analysis, decision to publish, or preparation of the manuscript."

"All authors are employees of H. Lundbeck A/S"

We note that one or more of the authors are employed by a commercial company: H. Lundbeck A/S.

6. Please include a caption for figure 1.

Reviewers' comments:

Reviewer's Responses to Questions

**Comments to the Author**

1. Is the manuscript technically sound, and do the data support the conclusions?

Reviewer #1: Partly

Reviewer #2: Yes

Reviewer #3: Yes

Reviewer #4: Partly

2. Has the statistical analysis been performed appropriately and rigorously? 

Reviewer #1: Yes

Reviewer #2: Yes

Reviewer #3: Yes

Reviewer #4: Yes

3. Have the authors made all data underlying the findings in their manuscript fully available?

Reviewer #1: No

Reviewer #2: No

Reviewer #4: No

4. Is the manuscript presented in an intelligible fashion and written in standard English?

Reviewer #1: Yes

Reviewer #2: Yes

Reviewer #3: Yes

Reviewer #4: Yes

5. Review Comments to the Author

Reviewer #1: I was unable to find the 'Supporting Information' files that the authors promised to provide, unless the intent was to consider the manuscript text itself as a 'Supporting Information' file.

In addition, the authors did not provide the spreadsheet or other software used to implement the Markov model.

These omissions actually make it difficult to make a full evaluation of this contribution, and some details required to confirm rigor and reproducibility of the estimates are missing.

That being said, if we make an assumption that the authors have acted and performed the research task with integrity and in a fashion that is acceptable from the standpoint of rigor and reproducibility, it should be noted that this contribution has a focus on the construct of 'Treatment-Resistant Schizophrenia,' often conceptualized as a categorical phenomenon or as one end of a spectrum of treatment responsiveness, and the size of the TRS caseload is of interest in pharmacoeconomics and pharmacoepidemiology as might be seen when pharmaceutical industry research and development decisions are made about investments in new medications for schizophrenia. Further, the results of the Markov modeling appear to be sound and logic supports descriptive statements provided by the authors, who offer several simulations, each based on reasonably plausible assumptions (e.g., concerning SMR to the extent that excess mortality might be shortening the lives of people who suffer from treatment-resistant schizophrenia. Further, there is a clearly stated set of limitations and assumptions, which should be helpful as a guide to future modeling work along these lines.

It is of interest that the authors adopted a deterministic approach and did not use simulation approaches that might provide interval estimates to complement the point estimates. (One might say that the three 'proxy' approaches lay out a set of alternative estimates that might be used in place of credible intervals or confidence intervals.)

A few minor comments:

The authors mention 'true prevalence' and this echoes some prior literature in the field, but indeed what we have are estimates and it might be argued that the construct of 'true prevalence' should have been abandoned long ago. The authors provide reasons for rejecting this notion, and they offer a range of estimates, each based on plausible assumptions about deterministic values in the Markov modeling simulations.

In general the paper is well-written and many PLOSONE readers should find it relatively easy to read and without a great deal of technical jargon. One exception involves a number of instances in which the subject of the sentence is a singular or a plural noun and the other parts of the sentence (e.g., verb form; subject complement) are the opposite. This situation can be remedied with some careful copyediting.

Reviewer #2: This study performs a Markov cohort simulation to model prevalence of treatment resistant schizophrenia. It uses data from public health databases as well as estimates from published estimates from systematic reviews.

Overall the study is interesting and contributes to the literature. The methods are appropriate to address the question, and it is written clearly and is readable for non-specialists. However, the report is also very brief and includes only one analysis using previously published data. Some recommendations are:

1) Various typos or other problems that should be corrected:

a. In the first paragraph of the introduction, line 5: ‘Some patients diagnosed with schizophrenia do only…’ Remove ‘do’.

b. Last sentence of the first paragraph of the introduction: ‘Additionally, half of the patients with TRS is…’ Remove ‘the’ and change ‘is’ to ‘are’.

c. Last sentence of the introduction: ‘United Stated’.

d. In the last paragraph of the Data section of the Methods (p.4, line 21): ‘TRS mortality was therefore adjusted proportionally adjusted…’ Repeated use of the word ‘adjusted’.

e. In the second paragraph of the Statistical Analysis section of the Methods (p.4, line 32): ‘The analysis were performed…’ Change ‘were’ to ‘was’.

f. The second sentence of the Discussion (‘The three different proxy definitions of TRS…’) is confusing. It would be helpful to reword it.

g. The last sentence of the first paragraph of the Discussion (‘albeit consistently higher estimates of TRS…’): United States is not capitalised properly.

h. First sentence of the second paragraph of the Discussion (‘Despite the one-way sensitivity analyses’): the sentence is confusing. Perhaps ‘revealed’ should be changed to ‘revealing’.

i. The last sentence of the third paragraph of the Discussion (‘the plausible ranges well coincide’) is clumsily worded. Perhaps ‘the plausible ranges coincide well’?

j. The last sentence of the Conclusion (‘The estimates are highly dependent of…’) is confusing. Perhaps ‘…are highly dependent on the proxy used…’?

2) If feasible to do so, it would be useful for the reader to include descriptives/summary statistics in the Results section about the data used for the analyses, although I am conscious that a number of data sets were used and that they were previously published elsewhere. At the moment it is unclear what the characteristics of the sample(s) used to estimate TMR prevalence were or how representative they were of the wider population.

3) In the Methods section, there are multiple assumptions described that were made for the model. For example, it is stated that the SMRs for TRS mortality was adjusted proportionally for proxy 1, but not for proxy 2 or 3. While I think that the rationale for these assumptions is reasonable, it would be helpful to add a discussion of these assumptions and how they may have impacted the results of the analysis to the limitations section in the Discussion.

Reviewer #3: I can congratulate the authors on their conceptualization of an interesting schizophrenia research question -- namely, how often treatment-resistant schizophrenia might be occurring in the United States. They appropriately identify a crucial issue- namely, how TRS case definitions are actualized.

The data they have pulled together seems quite appropriate, but it is possible that they have neglected some sources of data that are pertinent, and this topic deserves coverage in a revised limitations section of the paper.

Of special note might be the continuing uncertainty about whether any of the US databases now encompass the entire study population of cases of schizophrenia, within which the treatment-resistant case subset might be found.

Leaving aside issues of the completeness of the data and coverage of the US schizophrenia population, there are some uncertainties about the statistical modeling approach. Taking a step backward one might look for simulations that are more advanced (e.g., with Gibbs sampling), and here again, it is a topic for coverage in the Discussion section but not necessarily a serious impediment in relation to publication of this paper's estimates.

I remain unconvinced that the 22% estimate offered by the authors is that much different from the 30% estimate in the literature of the past. In addition, the description of the statistical approach does not mention how the 30% estimate might be used as a Bayesian prior, with the new evidence used to update that prior.

I'd be especially appreciative of a statistical approach that takes the 30% estimate as a Bayesian prior, and then evaluates that prior in relation to the new data now available.

Finally, I'd invite the authors to talk more about external generalizability, and here there is an important literature that might not be fully appreciated by PLOS readers and that stretches back to the US-UK diagnostic studies and the Sartorius-Wing group's work on the IPSS schizophrenia research. Some 38 citations are included, but I'm not seeing the important seminal work completed in that clearly pertinent study with samples from multiple countries.

Reviewer #4: This is an interesting paper based on simulations underpinned by epidemiological data to inform IPM models about the prevalence of TRS. The incidence data come from reliable sources, and the methodology appears sound, notwithstanding the points below which can mostly be easily addressed. I have a concern about the birth cohorts that were used for this exercise, and no method of validation of the model is reported in the methods, another major limitation.

1. Introduction: It’s unclear which reference the “30%” quote comes from. Is this based on a meta-analysis or similar?

2. Methods: Did you IPM model allow for the possibility of symptomatic or disease recovery / remission? If not, how could this have influenced your estimates?

3. Methods>Data: The model is restricted by other covariates over which incidence of schizophrenia varies, but which are not explicitly modelled. These would include ethnicity/migration status and cannabis use. Could the authors comment on how omitting these (and other) omitted variables could have affected TRS prevalence estimates?

4. Methods>Statistical Analysis: Please give more information about where you obtained the birth cohort data (1930-2014) from and how this was set up (i.e. stratified by sex?). How long were people simulated as at-risk of schizophrenia for? Typically this would be from ~16-64 years of age. I don’t understand the analysis fully here as you report the TRC prevalence for 2014. On this basis birth cohorts from 1998 onwards would not be relevant to your study, since they would be younger than 16 in 2014. The earliest lower age of onset I think you could go down to is 14 years, but 16 is more typical. This should be clarified. Equally, birth cohorts born in 1930 would be aged 84 in 2014, well passed the typical age of onset of psychosis. Very late onset psychosis has a distinct epidemiology, about which little is known with respect to TRS.

5. Methods>Statistical Analysis: How was the simulation model validated?

6. Table 1: Should there be corresponding confidence intervals around the prevalence of TRS and TRC proportion amongst people with schizophrenia?

7. Results: The text is unclear in places, for example “Comparing the one-way sensitivity analysis when changing the SMR rates…the model estimated an approximately 40% higher prevalence…” – it’s unclear to which line in Table 1 you are referring. Preference would be to report the TRS proportions specified in the Table, not the % change in TRS proportions. Similar argument for the “fivefold” increase.

Minor points:

8. Please remember to include page numbers which makes reviewing easier

9. Introduction: “…half of the patients with TRS is…” should be “…TRS are…”

10. Define acronyms on first usage, ICD, DSM etc

11. Table 1: You could indicate in the table, that Proxy 2 was the main definition to remind the reader at this point.

12. Discussion: Second sentence is unclear

13. Why did the authors chose the US population?

14. I did not have access to the full data in supplemental files, which were not provided to me. The authors may wish to check with PLoS One that open access to this data is available.

6. PLOS authors have the option to publish the peer review history of their article (what does this mean?). If published, this will include your full peer review and any attached files.

Reviewer #1: Yes: James C. Anthony

Reviewer #2: No

Reviewer #3: No

Reviewer #4: Yes: James B. Kirkbride Ph.D.

---

## [Author Response · Author response to Decision Letter 0]

6 Apr 2020

Author response to reviewers’ comments:

Reviewer #1: I was unable to find the 'Supporting Information' files that the authors promised to provide, unless the intent was to consider the manuscript text itself as a 'Supporting Information' file

In addition, the authors did not provide the spreadsheet or other software used to implement the Markov model.

These omissions actually make it difficult to make a full evaluation of this contribution, and some details required to confirm rigor and reproducibility of the estimates are missing.

Author response: Thanks for pointing this out to us. We think it is important that all data as well as the R-code used is available to the reviewers, and to the reader. In the resubmission we have provided the life tables for the life expectancy in the United States and the R code used in the IPM-model. In addition, we have made a Shiny-application in R, making it easy for the reader to change the proxies, SMRs etc, and see the resulting modelled prevalence of TRS in the United States. 

As for the SMRs, they come from published literature which we have cited.

That being said, if we make an assumption that the authors have acted and performed the research task with integrity and in a fashion that is acceptable from the standpoint of rigor and reproducibility, it should be noted that this contribution has a focus on the construct of 'Treatment-Resistant Schizophrenia,' often conceptualized as a categorical phenomenon or as one end of a spectrum of treatment responsiveness, and the size of the TRS caseload is of interest in pharmacoeconomics and pharmacoepidemiology as might be seen when pharmaceutical industry research and development decisions are made about investments in new medications for schizophrenia. Further, the results of the Markov modeling appear to be sound and logic supports descriptive statements provided by the authors, who offer several simulations, each based on reasonably plausible assumptions (e.g., concerning SMR to the extent that excess mortality might be shortening the lives of people who suffer from treatment-resistant schizophrenia. Further, there is a clearly stated set of limitations and assumptions, which should be helpful as a guide to future modeling work along these lines. It is of interest that the authors adopted a deterministic approach and did not use simulation approaches that might provide interval estimates to complement the point estimates. (One might say that the three 'proxy' approaches lay out a set of alternative estimates that might be used in place of credible intervals or confidence intervals.)

Author response: We thank the reviewer for the thoughtful comments, which we believe have improved the manuscript. 

A few minor comments:

The authors mention 'true prevalence' and this echoes some prior literature in the field, but indeed what we have are estimates and it might be argued that the construct of 'true prevalence' should have been abandoned long ago. The authors provide reasons for rejecting this notion, and they offer a range of estimates, each based on plausible assumptions about deterministic values in the Markov modeling simulations.

Author response: This is a valid point. In the revised version we do not refer to the true burden or true prevalence of TRS. 

In general the paper is well-written and many PLOSONE readers should find it relatively easy to read and without a great deal of technical jargon. One exception involves a number of instances in which the subject of the sentence is a singular or a plural noun and the other parts of the sentence (e.g., verb form; subject complement) are the opposite. This situation can be remedied with some careful copyediting.

Author response: We have had the manuscript thoroughly checked for such occasions. 

Reviewer #2: This study performs a Markov cohort simulation to model prevalence of treatment resistant schizophrenia. It uses data from public health databases as well as estimates from published estimates from systematic reviews.

Overall the study is interesting and contributes to the literature. The methods are appropriate to address the question, and it is written clearly and is readable for non-specialists. However, the report is also very brief and includes only one analysis using previously published data. Some recommendations are:

Author response: We would like to thank the reviewer for the good comments and suggestions, which we believe have helped improve the manuscript

1) Various typos or other problems that should be corrected:

a. In the first paragraph of the introduction, line 5: ‘Some patients diagnosed with schizophrenia do only…’ Remove ‘do’.

b. Last sentence of the first paragraph of the introduction: ‘Additionally, half of the patients with TRS is…’ Remove ‘the’ and change ‘is’ to ‘are’.

c. Last sentence of the introduction: ‘United Stated’.

d. In the last paragraph of the Data section of the Methods (p.4, line 21): ‘TRS mortality was therefore adjusted proportionally adjusted…’ Repeated use of the word ‘adjusted’.

e. In the second paragraph of the Statistical Analysis section of the Methods (p.4, line 32): ‘The analysis were performed…’ Change ‘were’ to ‘was’.

f. The second sentence of the Discussion (‘The three different proxy definitions of TRS…’) is confusing. It would be helpful to reword it.

g. The last sentence of the first paragraph of the Discussion (‘albeit consistently higher estimates of TRS…’): United States is not capitalised properly.

h. First sentence of the second paragraph of the Discussion (‘Despite the one-way sensitivity analyses’): the sentence is confusing. Perhaps ‘revealed’ should be changed to ‘revealing’.

i. The last sentence of the third paragraph of the Discussion (‘the plausible ranges well coincide’) is clumsily worded. Perhaps ‘the plausible ranges coincide well’?

j. The last sentence of the Conclusion (‘The estimates are highly dependent of…’) is confusing. Perhaps ‘…are highly dependent on the proxy used…’?

Author response: Thank you for these comments. All the comments above have been implemented in the revised version of the manuscript.

2) If feasible to do so, it would be useful for the reader to include descriptives/summary statistics in the Results section about the data used for the analyses, although I am conscious that a number of data sets were used and that they were previously published elsewhere. At the moment it is unclear what the characteristics of the sample(s) used to estimate TMR prevalence were or how representative they were of the wider population.

Author response: Thanks for pointing this out to us. As we wrote in the answer to reviewer 1, we think it is important that all data as well as the R-code used is available to the reviewers, and to the reader. 

We have provided the life tables for the life expectancy in the United States and the R code used in the IPM-model. In addition, we have made a Shiny-application in R, making it easy for the reader to change the proxies, SMRs etc, and see the resulting modelled prevalence of TRS in the United States. 

As for the SMRs, they come from published literature which we have cited.

3) In the Methods section, there are multiple assumptions described that were made for the model. For example, it is stated that the SMRs for TRS mortality was adjusted proportionally for proxy 1, but not for proxy 2 or 3. While I think that the rationale for these assumptions is reasonable, it would be helpful to add a discussion of these assumptions and how they may have impacted the results of the analysis to the limitations section in the Discussion.

Author response: This is a good suggestion. As additional analyses we adjusted the mortality risks for the other proxies downwards in a similar fashion to investigate the impact that would have to the results. This was also added in the discussion section. 

Reviewer #3: I can congratulate the authors on their conceptualization of an interesting schizophrenia research question -- namely, how often treatment-resistant schizophrenia might be occurring in the United States. They appropriately identify a crucial issue- namely, how TRS case definitions are actualized.

The data they have pulled together seems quite appropriate, but it is possible that they have neglected some sources of data that are pertinent, and this topic deserves coverage in a revised limitations section of the paper.

Author response: We would like to thank the reviewer for the suggestions and comments on our work. These helped improve the paper. As suggested, we have added the impact of possibly omitting some sources in the limitations section.

Of special note might be the continuing uncertainty about whether any of the US databases now encompass the entire study population of cases of schizophrenia, within which the treatment-resistant case subset might be found. 

Author response: We unfortunately don’t have access to such US database with a complete schizophrenic population. Applying the same proxies to estimate the prevalence of TRS in such a setting would be very interesting. Our approach, using the basic code which is now made available to the reader, should however be transferable to other countries or regions without the need to access a database. That is, by uploading region or country specific life tables the reader should be able to generate estimates of the TRS prevalence rapidly.

Leaving aside issues of the completeness of the data and coverage of the US schizophrenia population, there are some uncertainties about the statistical modeling approach. Taking a step backward one might look for simulations that are more advanced (e.g., with Gibbs sampling), and here again, it is a topic for coverage in the Discussion section but not necessarily a serious impediment in relation to publication of this paper's estimates.

Author response: We agree with the reviewer that it would be of interest to perform a more advanced simulation to obtain uncertainty around the estimates. E.g. sampling 10.000 times from the distribution in a frequentist manner or using a Gibbs sampling approach in a bayesian framework. The current deterministic sensitivity analysis explores the one-way uncertainty in the estimate, and we like the simplicity of the model, which makes it easy for the reader to understand the model and how changes in the parameters impact the prevalence.

I remain unconvinced that the 22% estimate offered by the authors is that much different from the 30% estimate in the literature of the past. In addition, the description of the statistical approach does not mention how the 30% estimate might be used as a Bayesian prior, with the new evidence used to update that prior. I'd be especially appreciative of a statistical approach that takes the 30% estimate as a Bayesian prior, and then evaluates that prior in relation to the new data now available.

Author response: We agree with the reviewer that the difference is not that large, and we have worded this carefully where we state that “The results suggests that prevalence of TRS may be somewhat lower than the 30% often reported, however this is highly dependent on the definition of treatment resistance”

Finally, I'd invite the authors to talk more about external generalizability, and here there is an important literature that might not be fully appreciated by PLOS readers and that stretches back to the US-UK diagnostic studies and the Sartorius-Wing group's work on the IPSS schizophrenia research. Some 38 citations are included, but I'm not seeing the important seminal work completed in that clearly pertinent study with samples from multiple countries.

Author response: We would like to thank the reviewer for pointing out this literature. To the external generalizability, an extension to the limitations section have been added to accommodate this.

Reviewer #4: This is an interesting paper based on simulations underpinned by epidemiological data to inform IPM models about the prevalence of TRS. The incidence data come from reliable sources, and the methodology appears sound, notwithstanding the points below which can mostly be easily addressed. I have a concern about the birth cohorts that were used for this exercise, and no method of validation of the model is reported in the methods, another major limitation.

Author response: Thank you for your constructive comments on our paper. 

1. Introduction: It’s unclear which reference the “30%” quote comes from. Is this based on a meta-analysis or similar?

Author response: We have added references in the revised version of the paper.

2. Methods: Did you IPM model allow for the possibility of symptomatic or disease recovery / remission? If not, how could this have influenced your estimates?

Author response: This is an important point and we state in the method section that the model did not consider the possibility of disease recovery. We have added a discussion on this in the limitation section.

3. Methods>Data: The model is restricted by other covariates over which incidence of schizophrenia varies, but which are not explicitly modelled. These would include ethnicity/migration status and cannabis use. Could the authors comment on how omitting these (and other) omitted variables could have affected TRS prevalence estimates?

Author response: This is a good suggestion. We have added a paragraph on this in the limitation section. 

4. Methods>Statistical Analysis: Please give more information about where you obtained the birth cohort data (1930-2014) from and how this was set up (i.e. stratified by sex?). How long were people simulated as at-risk of schizophrenia for? Typically this would be from ~16-64 years of age. I don’t understand the analysis fully here as you report the TRC prevalence for 2014. On this basis birth cohorts from 1998 onwards would not be relevant to your study, since they would be younger than 16 in 2014. The earliest lower age of onset I think you could go down to is 14 years, but 16 is more typical. This should be clarified. Equally, birth cohorts born in 1930 would be aged 84 in 2014, well passed the typical age of onset of psychosis. Very late onset psychosis has a distinct epidemiology, about which little is known with respect to TRS.

Author response: Good point. We set the transition probabilities for the population under 15 years of age or above 70 years of age between the population and schizophrenia state to be 0. Therefore, it was not possible for those age groups to become schizophrenics. We have added this explanation in the revised version of the manuscript.

National statistics life tables were provided by the U.S Social Security Administration and can be found in the supplementary material.

5. Methods>Statistical Analysis: How was the simulation model validated?

Author response: This is an important point. We internally validated the model and its methodological framework among clinical and statistical experts in Lundbeck to an extend where our experts believed it was both clinical and statistical sound, however, this is not discussed in the article. Additionally, we discuss confounding and generalizability of the model and its framework in the discussion and limitations section.

6. Table 1: Should there be corresponding confidence intervals around the prevalence of TRS and TRC proportion amongst people with schizophrenia?

Author response: We think that the intervals provided in the table are highlighting the uncertainties in the estimates. 

7. Results: The text is unclear in places, for example “Comparing the one-way sensitivity analysis when changing the SMR rates…the model estimated an approximately 40% higher prevalence…” – it’s unclear to which line in Table 1 you are referring. Preference would be to report the TRS proportions specified in the Table, not the % change in TRS proportions. Similar argument for the “fivefold” increase.

Author response: We have changed this according to the reviewer’s suggestion.

Minor points:

8. Please remember to include page numbers which makes reviewing easier

Author response: We have added this.

9. Introduction: “…half of the patients with TRS is…” should be “…TRS are…”

Author response: Changed

10. Define acronyms on first usage, ICD, DSM etc

Author response: Thanks for pointing this out to us. We have gone through the manuscript again and corrected all such occasions.

11. Table 1: You could indicate in the table, that Proxy 2 was the main definition to remind the reader at this point.

Author response: Good suggestion, we added this in the table.

12. Discussion: Second sentence is unclear

Author response:

13. Why did the authors chose the US population?

Author response: We chose the US population because of the available public data for this population and due to the size of the patient population and its burden on society.

14. I did not have access to the full data in supplemental files, which were not provided to me. The authors may wish to check with PLoS One that open access to this data is available.

Author response: Thanks for pointing this out to us. These should be available in the resubmission.

---

## [Decision Letter · Decision Letter 1]

8 May 2020

PONE-D-19-32237R1

A modelling approach to estimate the prevalence of treatment-resistant schizophrenia in the United States

PLOS ONE

Dear Mr. Mørup,

Thank you for submitting your manuscript to PLOS ONE. After careful consideration, we feel that it has merit but does not fully meet PLOS ONE’s publication criteria as it currently stands. Therefore, we invite you to submit a revised version of the manuscript that addresses the points raised during the review process.

We would appreciate receiving your revised manuscript by Jun 22 2020 11:59PM. To enhance the reproducibility of your results, we recommend that if applicable you deposit your laboratory protocols in protocols.io, where a protocol can be assigned its own identifier (DOI) such that it can be cited independently in the future. For instructions see: http://journals.plos.org/plosone/s/submission-guidelines#loc-laboratory-protocols

We look forward to receiving your revised manuscript.

Kind regards,

Giuseppe Carrà, PhD

Academic Editor

PLOS ONE

Reviewers' comments:

Reviewer's Responses to Questions

**Comments to the Author**

1. If the authors have adequately addressed your comments raised in a previous round of review and you feel that this manuscript is now acceptable for publication, you may indicate that here to bypass the “Comments to the Author” section, enter your conflict of interest statement in the “Confidential to Editor” section, and submit your "Accept" recommendation.

Reviewer #2: All comments have been addressed

Reviewer #4: (No Response)

2. Is the manuscript technically sound, and do the data support the conclusions?

Reviewer #2: Partly

Reviewer #4: Yes

3. Has the statistical analysis been performed appropriately and rigorously? 

Reviewer #2: Yes

Reviewer #4: Yes

4. Have the authors made all data underlying the findings in their manuscript fully available?

Reviewer #2: Yes

Reviewer #4: Yes

5. Is the manuscript presented in an intelligible fashion and written in standard English?

Reviewer #2: Yes

Reviewer #4: Yes

6. Review Comments to the Author

Reviewer #2: The authors have largely addressed the issues raised and the paper is improved for it. One final suggestion would be to add a limitation to the paper clearly acknowledging the potential issues with data accuracy and reliability in the datasets they have used.

Reviewer #4: Thank you for addressing my concerns, which are generally clear and have helped to improve the manuscript. One concern that I feels requires further detail and revision is over validation. The abstract of the paper states that the model was validated for the purpose of estimate the prevalence of TRS in the US. The author response to my earlier concern with this issue, however, noted that validity had been established by asking clinicians and statistical experts about the model and its methodological framework. This is not really internal validity but some measure of apparent or face validity. Internal validity would be more formally established using bootstrapping or some other cross-validation procedure applied to the available data. There is also no mention of any external validation (do the predictions from your model accurately predict TRS prevalence in the US in a representative sample?), which may not be possible in the absence of a reliable external dataset in which to test this. If so, external validation can simply be acknowledged as a limitation, but I would think there is more the authors could do to internally validate their models.

7. PLOS authors have the option to publish the peer review history of their article (what does this mean?). If published, this will include your full peer review and any attached files.

Reviewer #2: No

Reviewer #4: Yes: James B. Kirkbride

---

## [Author Response · Author response to Decision Letter 1]

18 May 2020

Author response to reviewers’ comments:

Reviewer #2: The authors have largely addressed the issues raised and the paper is improved for it. One final suggestion would be to add a limitation to the paper clearly acknowledging the potential issues with data accuracy and reliability in the datasets they have used.

Author response: We would like to thank the reviewer for the thoughtful comments on this and the previous version of the manuscript.

We added a paragraph in the limitation section focusing on potential issues with the data used to model TRS.

Reviewer #4: Thank you for addressing my concerns, which are generally clear and have helped to improve the manuscript. One concern that I feels requires further detail and revision is over validation. The abstract of the paper states that the model was validated for the purpose of estimate the prevalence of TRS in the US. The author response to my earlier concern with this issue, however, noted that validity had been established by asking clinicians and statistical experts about the model and its methodological framework. This is not really internal validity but some measure of apparent or face validity. Internal validity would be more formally established using bootstrapping or some other cross-validation procedure applied to the available data. There is also no mention of any external validation (do the predictions from your model accurately predict TRS prevalence in the US in a representative sample?), which may not be possible in the absence of a reliable external dataset in which to test this. If so, external validation can simply be acknowledged as a limitation, but I would think there is more the authors could do to internally validate their models.

Author response: We thank the reviewer for these thoughtful comments and for helping us improve the quality of the paper.

The reviewer raises an important point about the validation of the model. We deleted the statement in the abstract regarding validation, as the model, as noted by the reviewer, has not been thoroughly validated. We did expand the limitation section to address issues concerning external validation.

We do however disagree with the reviewer that we can do more regarding internal validity. The reviewer suggested more could be done using on internally validating the model by use of bootstrapping or cross-validation. Such approaches are very useful when data allows sampling, however the data used in the present simulation does not allow for such sampling. The life tables used to derive the age, gender and cohort specific mortality risks does not have any variation in it, nor does the population data. Regarding the epidemiological inputs, the base scenario for each proxy were based on the medians from published literature and were presented along with the 10th and 90th percentiles, which produced a deterministic range for the results.

---

## [Editor Report · Decision Letter 2]

20 May 2020

A modelling approach to estimate the prevalence of treatment-resistant schizophrenia in the United States

PONE-D-19-32237R2

Dear Dr. Mørup,

We are pleased to inform you that your manuscript has been judged scientifically suitable for publication and will be formally accepted for publication once it complies with all outstanding technical requirements.

With kind regards,

Giuseppe Carrà, MD, PhD

PLOS ONE Academic Editor

---

## [Editor Report · Acceptance letter]

27 May 2020

PONE-D-19-32237R2 

A modelling approach to estimate the prevalence of treatment-resistant schizophrenia in the United States 

Dear Dr. Mørup:

I am pleased to inform you that your manuscript has been deemed suitable for publication in PLOS ONE. Congratulations! Your manuscript is now with our production department. 

With kind regards,

on behalf of

Dr. Giuseppe Carrà 

Academic Editor

PLOS ONE